# Pharmacokinetics of Four Tyrosine Kinase Inhibitors in Adult and Paediatric Chronic Myeloid Leukaemia Patients

**DOI:** 10.3390/biomedicines11092478

**Published:** 2023-09-07

**Authors:** Sarah Allegra, Emma Dondi, Francesco Chiara, Silvia De Francia

**Affiliations:** Laboratory of Clinical Pharmacology “Franco Ghezzo”, Department of Clinical and Biological Sciences, University of Turin, S. Luigi Gonzaga Hospital, 10043 Orbassano, TO, Italy; emma.dondi@unito.it (E.D.); 336124@edu.unito.it (F.C.); silvia.defrancia@unito.it (S.D.F.)

**Keywords:** dasatinib, imatinib, nilotibin, ponatinib, children, therapeutic drug monitoring, sex and gender pharmacology, personalized medicine

## Abstract

Tyrosine kinase inhibitors work by blocking the tyrosine kinases responsible for the dysregulation of intracellular signalling pathways in tumour cells. This study looked at the impact of age and sex on the levels of imatinib, dasatinib, nilotinib, and ponatinib in plasma and cerebrospinal fluid samples of patients with chronic myeloid leukaemia. Imatinib and dasatinib were used to treat the majority of the enrolled patients, and most of them were paediatrics. A total of 82.4% of the patients were men; however, sex-related differences in the drugs’ pharmacokinetics were not found. Age and imatinib plasma concentration were found to be inversely correlated. The dasatinib concentrations in plasma were found to be substantially lower than those found in cerebrospinal fluid, particularly in paediatrics. Analysing the obtained data, we can state that therapeutic drug monitoring is a useful method for adjusting a patient’s treatment schedule that depends on drug concentrations in biological fluids. The use of therapeutic drug monitoring in conjunction with tyrosine kinase inhibitors for the treatment of chronic myeloid leukaemia is supported by a number of sources of evidence. As a result, as the research develops, the tyrosine kinase inhibitor therapeutic drug monitoring classification needs to be refined in terms of factors like sex and age.

## 1. Introduction

Chronic myeloid leukaemia (CML) is a clonal stem cell disorder. The WHO classification of 2008 places it within the BCR–ABL1 positive myeloproliferative neoplasms [1]. It is a relatively rare neoplastic disease with an incidence of approximately 1–2 cases/100,000 inhabitants/year. The median age is approximately 56 years, and approximately 30% of patients present with an age >60 years; however, there are juvenile-onset cases [2].

The pathogenesis of the disease is due to a clonal anomaly of the stem cell, which manifests itself with an excessive proliferation of myeloid precursors in the bone marrow due to an acquired chromosomal anomaly, the Philadelphia chromosome (Ph’), that is present in 95% of cases of CML. The Ph’ chromosome is the result of the balanced translocation between the Abelson gene (ABL1) from chromosome 9 and the region of chromosome 22 called the breakpoint cluster region (BCR). Therefore, the result of the t(9;22) (q34;q11.2) translocation is the BCR–ABL1 fusion protein with a tyrosine kinase activity inherited from ABL1 that is constitutive and, therefore, no longer regulated. Two different forms of the protein are known depending on its molecular weight, p210 (b3a2, b2a2) and p190 (e1a2). BCR–ABL1 is a cytoplasmic tyrosine kinase that determines an alteration of the proliferation, apoptosis, and cell adhesion of neoplastic cells. The leukemic clone gradually replaces the healthy cells involved [3].

In cases of CML in which the Ph’ chromosome is not observed, which are approximately 5% of the total, the presence of the BCR–ABL1 fusion gene can be demonstrated with molecular biology techniques. The presence of this molecular alteration distinguishes CML from Ph’ negative (Ph−) myeloproliferative diseases. There are rare cases of neutrophilic leukaemia and Ph− CML. The clinical picture of CML mainly consists of splenomegaly, leucocytosis, and platelets, sometimes even of considerable entity, with or without anaemia, which is associated with a corollary of non-specific symptoms, such as asthenia, abdominal tension, an early sense of fullness, arthralgias, myalgias, and in some cases, weight loss [4]. If left untreated, it is characterized by a fatal clinical course within a few years.

In the initial phase of the disease, known as the chronic phase, the leukaemia cells tend to grow in number with an increase in the number of white blood cells in the peripheral blood at different stages of maturation and in the volume of the spleen, but they retain the ability to mature and produce blood cells. During this stage, disease control is quite easy. After a variable period of time, even a few years, a second phase follows called the accelerated phase, generally lasting months or a few years, in which the disease becomes more aggressive and, if not treated, inevitably evolves into the blast phase in which the cancer cells lose the ability to mature and the disease takes on the characteristics of acute leukaemia [5].

The history of CML has changed dramatically thanks to the introduction in the 1990’s of tyrosine kinase inhibitors, which are drugs capable of blocking the constitutive activation of BCR–ABL1. Treatment requires drug monitoring (therapeutic drug monitoring—TDM) in order to improve the effectiveness of cancer therapy and allow for the customization of doses, to evaluate patient adherence, to identify potential drug interactions, to assess treatment efficacy, and to limit side effects [6,7].

Imatinib, commercially known as Glivec^®^, is a drug designed for the treatment of CML, acute myeloid leukaemia, and gastrointestinal cancer. In CML, it is indicated for the treatment of newly diagnosed Ph+ adult and paediatric patients for whom bone marrow transplantation is not considered as the first-line treatment, for Ph+ adult and paediatric patients in the chronic phase after failure of interferon-alpha therapy, and for Ph+ adult and paediatric patients in the accelerated phase or blast crisis. The recommended daily dose is 400–800 mg to be given orally. Imatinib bioavailability is almost complete (98%) and does not depend on food assumption. The drug absorption in the gastrointestinal tract is quick (1–2 h after the administration with a plasma peak after 2–4 h), and the distribution is mostly performed by bonds with plasma proteins. The imatinib metabolism is hepatic, and the excretion is biliary and urinary. Imatinib’s half-life is approximately 18 h [8]. The development of resistance is a relevant limit to the therapy with imatinib, which is why new and more efficient drugs, such as dasatinib and nilotinib, have been developed [9].

Dasatinib is commercially known as Spycel^®^. It is indicated for the treatment of adult patients with newly diagnosed Ph+ CML; CML in the chronic phase, accelerated phase, or blast phase with resistance or intolerance to prior therapy including imatinib; Ph+ acute lymphoblastic leukaemia (ALL), and lymphoid blast-phase CML with resistance or intolerance to prior therapy. In addition, it is recommended for the treatment of paediatric patients with newly diagnosed Ph+ CML or Ph+ chronic phase CML with resistance or intolerance to prior therapy including imatinib and in newly diagnosed Ph+ ALL in combination with chemotherapy. The initial oral dose in chronic phase patients is 100 mg/day, while in the accelerated phase, in the myeloid blastic and lymphoid leukaemia, and in the acute lymphoblastic leukaemia, the oral dose is 70 mg (1 tab in the morning and 1 in the evening). Dasatinib is rapidly absorbed with a concentration peak in 0.5–3 h after the administration. Dasatinib presents a high distribution volume (>3 L/kg) performed by bonds with plasma proteins; the oral availability varies between 14 and 34%, and the average half-life is approximately 1.3–5 h. Drug metabolism is hepatic, and the excretion is faecal and renal [10].

Nilotinib, commercially known as Tasigna^®^, like imatinib and dasatinib, is a BCR–-ABL kinase selective inhibitor that is capable of inhibiting proliferation and inducting apoptosis. Nilotinib is indicated for the treatment of newly diagnosed Ph+ CML adult and paediatric patients in the chronic phase, Ph+ CML adult patients in the chronic phase and accelerated phase with resistance or intolerance to prior therapy including imatinib, and Ph+ CML paediatric patients in the chronic phase with resistance or intolerance to prior therapy including imatinib. As does imatinib, nilotinib inhibits the tyrosine kinase protein’s activity but with a strength 20–50 times higher than imatinib. A total of 400 mg of nilotinib has to be taken twice a day without food assumption for at least 2 h before the intake and 1 h after it; bioavailability is in fact increased through food assumption. Nilotinib’s bioavailability is approximately 30% with a concentration peak in 3 h after the administration. The nilotinib metabolism is hepatic, the excretion is faecal and biliary, and the average half-life is approximately 15 h [11].

Ponatinib, commercially known as Iclusig^®^, is a multi-target kinase inhibitor, and its primary cellular target is BCR–ABL. It is indicated in patients with chronic, accelerated, or blast phase CML who are resistant or intolerant to dasatinib or nilotinib and for whom subsequent treatment with imatinib is not clinically appropriate or in whom the T315I mutation has been identified, and in patients with Ph+ ALL who are resistant or intolerant to dasatinib and for whom further treatment with imatinib is not clinically appropriate or in whom the T315I mutation has been identified. The recommended dose is 45 mg once daily for 28 days, and food assumption does not affect the absorption. The half-life is approximately 24 h, and the peak concentration is observed within 6 h after the administration. Ponatinib is metabolized through multiple pathways and eliminated mostly in faeces [12].

The aim of this study was to examine the influence of age and sex on the plasma and cerebrospinal fluid (CSF) concentrations of four different TKIs (imatinib, dasatinib, nilotinib, and ponatinib) in patients with CML.

## 2. Materials and Methods

### 2.1. Patients

We performed a monocentric cohort study of adult and paediatric patients with CML for whom plasma and CSF monitoring were required at the Clinical Pharmacology Service “Franco Ghezzo” of the San Luigi Gonzaga University Hospital in Orbassano (Turin, Italy) from January 2017 to December 2022. The inclusion criteria were CML diagnosis at an age above 2 years and treated with dasatinib, imatinib, nilotinib, or ponatinib for at least 6 months with a self-reported adherence of 90%. After obtaining their informed consent, the patients with CML who received regular doses of imatinib, dasatinib, ponatinib, or nilotinib underwent blood and, if deemed appropriate by the attending physician, CSF sampling for the purpose of measuring the drug concentrations at the end of the dosing interval (Cthrough). In particular, CSF sampling was carried out only in patients with central nervous system involvement. Ethics committee approval was not required, but the research project was the same one submitted to the local ethics committee (Prot. N° 2002, approved). Confidentiality was guaranteed in the data collection, analysis, and dissemination phases by presenting the results in aggregate form. The collected data were age, sex, CML diagnosis, and TKI treatments.

### 2.2. Chromatographic System and Conditions

The chromatographic analysis for the intracellular quantification of the four drugs was carried out using an HPLC–UV system. HPLC was performed with the Agilent 1100 system (Santa Clara, CA, USA) equipped with an autosampler, a spectrophotometer, and a heated column compartment. System management and data acquisition were performed with the ChemStation Agilent software REV. A 09.03 (Santa Clara, CA, USA). The chromatographic separation was realized at 40 °C using a HyPURITY C18 (ThermoScientific, Monza, Italy) 150 × 4.6 mm 3 µ column. The analysis was carried out at the constant flow rate of 0.9 mL/min at 35 °C in an isocratic condition. The eluate was monitored at 267 nm. The time of the analytical run was 10 min, according to the retention times of the substances and their good separation.

### 2.3. Chemicals

Imatinib, nilotinib, dasatinib, ponatinib, triethylamine, methanol, and acetonitrile were purchased from Sigma-Aldrich (Milan, Italy). A Milli DI system from Millipore (Milan, Italy) was used to create high pressure liquid chromatography (HPLC)-grade water. For the chromatographic run, a mobile phase constituted by 40% of solution A (72.5% H_2_O + 25% CH_3_OH + 2.5% C_6_H_15_N), 40% C_2_H_3_N, and 20% CH_3_OH was used.

### 2.4. Stock Solutions, Calibration Standards (STDs), Quality Controls (QCs), and Plasma Patient Samples

The Blood Bank of S San Luigi Gonzaga University Hospital in Orbassano (Turin, Italy) generously provided blank plasma from healthy donors that was used to prepare the STDs and QCs. The drug stock solutions were created in methanol at a final concentration of 1 mg/mL and kept at −4 °C for no longer than three months. The internal standard (IS) for the dasatinib, imatinib, and ponatinib evaluations was nilotinib, whereas for the nilotinib evaluation, the IS was imatinib. The 50 g/mL methanol-based IS solutions were prepared and utilized right away. The highest calibration standard (STD8) and the highest quality control (QC5) were obtained by diluting donors’ blank plasma serially with the other STDs and QCs, and the highest calibration standard (STD8) and QC5 were obtained by adding a specific volume of stock solutions to blank plasma. All the drugs used the same calibration range and QC concentrations (STDs: 0.005, 0.01, 0.05, 0.1, 0.5, 1, 5, 10 µg/mL; QCs: 0.05, 0.5, 5 µg/mL). Prior to analysis, the STDs and QCs were kept at −4 °C for no more than three months, avoiding multiple freeze–thaw cycles. Calibration curves and QCs were routinely recorded during daily clinical practice as a quality assurance measure and in order to explore improvements in the quality of services. The limit of detection (LOD) in plasma was defined as the concentration referred to a signal-to-noise ratio of 3/1; the lowest concentration levels determined with a percent of deviation from the nominal concentration and a relative standard deviation < 20% was considered the lowest limit of quantification (LOQ).

Plasma samples from patients with CML who were undergoing imatinib, dasatinib, nilotinib, or ponatinib treatment were obtained after the blood samples were centrifuged at 1500 g for 10 min at 4 °C, and the samples were kept at −4 °C until HPLC coupled with ultraviolet detection (UV) analysis was performed. CSF samples, collected by an experienced physician in the lumbar region of patients with CML who were undergoing imatinib, dasatinib, nilotinib, or ponatinib treatment, were directly frozen until HPLC–UV analysis was performed.

### 2.5. Sample Preparation

The patients’ plasma and CSF samples, STDs, and QCs were all treated with the following procedure. First, for dasatinib and ponatib, 50 µL of IS was added to 500 µL of sample. Then, 500 µL of C_2_H_3_N was used for the deproteinization and vortexed for 30 s before carrying out the centrifugation (15′ at 12.000 rpm). The following steps need the usage of a void pump: C18 Solid Phase Extraction (SPE; Grace, Italy) columns were conditioned with 1 mL of CH_3_OH and 1 mL of H_2_O; and a total of 800 µL of the obtained sample supernatant was transferred onto the SPE columns with 1 mL of H_2_O. After removing the waste, 500 µL of CH_3_OH was added; therefore, the solution needed to evaporate until a pellet was obtained. Finally, 250 µL of the mobile phase was added; the solution was then ready for the HPLC–UV analysis.

Differently, for nilotinib and imatinib, 50 µL of IS was added to 500 µL of the plasma or CSF sample. Then, 750 µL of extracting solution (C_2_H_3_N:CH_3_OH 50:50) was added and vortexed for 30 s before carrying out the centrifugation (10′ at 12.000 rpm). The solution was then ready for the HPLC–UV analysis.

### 2.6. Statistical Analysis

The one-way ANOVA test was used to calculate the power analysis. To carry out the descriptive analysis, the continuous and non-normal variables were summarized as the median and the interquartile range (IQR; quartile 1; quartile 3) to evaluate the statistical dispersion of the data; the categorical variables were represented as the frequency and percentage. All the variables were tested for normality with the Shapiro–Wilk test. The correspondence of each parameter was evaluated with a normal or non-normal distribution through the Kolmogorov–Smirnov test. Firth’s correction was applied to reduce the bias of the estimates due to the small number of events. The differences between males and females were tested with the Mann–Whitney U test or the Fisher exact test, when appropriate. The Pearson linear correlation coefficient (r) was used to investigate the strength of the association between two quantitative variables. Since the analysis was carried out at visit-level, the Huber–White estimator was used to adjust the correlation between multiple observations on the same patient. Odds ratios (OR) and their 95% confidence intervals (95% CI) were reported. Firth’s correction was applied to reduce the bias of the estimates due to the small number of events. All the tests were performed with IBM SPSS Statistics 25.0 for Windows (Chicago, IL, USA). The level of significance was set at 0.05.

## 3. Results

### 3.1. Study Population

We enrolled 153 patients diagnosed with CML. The one-way ANOVA test, calculated considering the different groups of samples (imatinib, dasatinib, nilotinib, and ponatinib), reported a *p*-value <0.001. The flow chart of the patient’s distribution is shown in Figure 1.

The median age of the enrolled patients was 13 years, and most of them were males (N = 126 males versus 27 females) and paediatrics (N = 116 paediatrics versus 37 adults). The demographics and pharmacological characteristics are reported in Table 1 stratified by the drug assumed and age (paediatrics and adults).

### 3.2. Imatinib Pharmacokinetics

The median age of the 60 patients undergoing imatinib therapy was 15 years (IQR 25–56.75 years), and 31 patients (51.7%) were paediatrics. The imatinib median plasma concentration was 1.82 µg/mL (IQR 0.88–2.93 µg/mL), and CSF sampling was not performed on any of the included patients. A borderline correlation between imatinib plasma concentration and age was observed: *p* = 0.073, r = −0.233. No significant influence of sex and age groups (paediatrics versus adults) was observed with the Mann–Whitney test. No statistically significant results were observed considering paediatrics and adults separately.

### 3.3. Dasatinib Pharmacokinetics

Seventy-seven patients were treated with dasatinib, and they were all men. The median age was 13 years (IQR 12–17 years), and 72 patients (93.5%) were paediatrics. The median plasma concentration collected in 39 patients was 0.06 µg/mL (IQR 0.03–0.33 µg/mL); the median CSF level collected in 38 patients was 0.11 µg/mL (IQR 0.06–0.32 µg/mL). Considering plasma and CSF matrices, a statistically significant influence was observed on drug concentrations (*p* = 0.038): plasma levels (N = 39; median 0.06 µg/mL, IQR 0.03–0.33 µg/mL) are lower than CSF levels (N = 38; median 0.11 µg/mL, IQR 0.06–0.32 µg/mL) as shown in Figure 2.

With the Mann–Whitney test, age groups significantly influence dasatinib plasma concentrations (*p* = 0.022): in paediatrics (N = 36; median 0.05 µg/mL, IQR 0.27–0.13 µg/mL), the drug concentrations are higher than those in adults (N = 3; median 0.4 µg/mL, IQR 0.28–0.74 µg/mL) as shown in Figure 3.

Considering the different pharmacokinetic distribution in paediatrics and adults, we decided to separately evaluate the two groups. We observed that the significant difference between plasma and CSF dasatinib levels was retained only in paediatrics: plasma levels (N = 36; median 0.05 µg/mL, IQR 0.03–0.13 µg/mL) are lower than CSF levels (N = 36; median 0.11 µg/mL, IQR 0.05–0.29 µg/mL) as shown in Figure 4.

The Pearson correlation showed no significant results between the considered variables.

### 3.4. Nilotinib and Ponatinib Pharmacokinetics

Only two of the enrolled patients were treated with nilotinib, and both of them were males. Their median age was 52 years (IQR 50–54 years); the mean plasma concentration was 0.19 μg/mL (IQR 0.16–0.22 μg/mL). CSF sampling was not performed on any of the patients. Considering ponatinib, 14 patients were enrolled, and also in this case, they were all males. Paediatrics were 13 (32.9%), and only 1 (7.1%) was an adult. The median plasma ponatinib concentration was 0.19 μg/mL (IQR 0.15–0.51 μg/mL), and the median CSF ponatinib concentration was 0.65 μg/mL (IQR 0.18–1.65 μg/mL). With the performed statistical analyses, no factors significantly influenced or were correlated with nilotinib and ponatinib concentrations.

## 4. Discussion

Patients with CML at diagnosis are classified into three risk categories, high, intermediate, and low risk, according to the Sokal and Hasford, European Treatment and Outcome Study for CML (EUTOS), and EUTOS long-term survival (ELTS) risks [13,14,15]. The Sokal and Hasford risks assess the risk of disease evolution, the EUTOS risk takes more into account the probability of obtaining a CCyR at 18 months, and the ELTS risk calculates the probability of dying from CML. All these scores consider, for the prediction of low, intermediate, or high risk, variables such as age, spleen size, platelet count, and percent of blasts, eosinophils, and basophils of peripheral blood. In clinical practice, currently, the Sokal risk is the most used; however, the ELTS score is more predictive and, therefore, evaluation according to this score is also suggested.

The recommendations of the European Leukaemia Net (ELN) published in 2013 by Baccarani et al. define what are the response criteria to the therapy and the times to reach it [16]. They also define how to use TKIs in the various lines of therapy. The presence of comorbidities, carefully evaluated, suggests caution in the use of the following:Imatinib in cases of renal insufficiency or uncontrolled heart failureNilotinib in cases of relevant ischemic cardiovascular disease and poorly controlled diabetes mellitusDasatinib in cases of relevant respiratory pathologies, pulmonary hypertension, and gastrointestinal bleedingBosutinib in cases of renal insufficiencyPonatinib in cases of cardiovascular pathologies.

Drug doses can be adjusted on the basis of efficacy and tolerance and/or toxicity.

With regard to the management of toxicities during therapy, please refer to the NCCN, ELN guidelines and the technical data sheets of the individual drugs.

As shown in Figure 1, the majority of patients included in our study were treated with imatinib and dasatinib. Despite the observational study drawbacks, it was found that the nilotinib and dasatinib outcomes of the New TARGET observational study 1 were significantly better than those of imatinib [17]. Contrarily, second generations are typically thought to cause more severe side effects than imatinib, including cardiovascular complications. Thus, comorbidities and side effects should be carefully considered while choosing the TKI. Moreover, in cases of vascular illness, nilotinib and ponatinib are contraindicated for the treatment of CML [12,18,19,20]; imatinib is still the first-line treatment that is advised in these situations. In addition, TKI treatment choices depend in part on the patients’ ages. This target therapy may be stopped in younger individuals if a full molecular remission is achieved permanently. In this case, second-generation TKIs might be preferable over imatinib due to their higher rate of full molecular remission. On the other hand, older people are unaffected when TKI medication is stopped [21]. In our cohort, most of the patients were under 18 years of age. Particularly, we observed that the distribution of the imatinib adult and paediatric population is similar. Whereas, for dasatinib, the majority of included patients were paediatrics. This disparity could be due to dasatinib-related pleural effusion observed in a great percentage of newly diagnosed and imatinib-resistant/intolerant adults with CML; moreover, dasatinib-related pulmonary hypertension was recorded in 5% of newly diagnosed adults with CML and in 2% of adults with CML who are imatinib-resistant/intolerant [22,23]. Probably, paediatrics in our study had less or no cardiopulmonary comorbidities that candidate them to dasatinib treatment. Considering nilotinib and ponatinib, the total number of patients included in the study is too small to try to explain the reason for their distribution.

The sex pharmacology approach in a wide field is often very difficult to pursue. It is yet unclear how sex differences, particularly those resulting from sex-specific immunological responses to CML, can affect clinical outcomes. As reported in the literature, men are more likely than women to develop CML with the M:F ratio ranging between 1.2 and 1.7 [24,25,26]. In younger age groups, the incidence of the sex difference is less pronounced. This sex-related incidence is confirmed in our study where 82.4% of the enrolled patients were males. Evaluating drug pharmacokinetics, we did not observe sex-related differences. In imatinib pharmacokinetic investigations, no sex differences have been observed [27]; also, treatment-related toxicity and quality of life improvement do not seem to be sex-specific [28]. Women appear to have a better outcome or at least a comparable outcome when presenting with less-favourable prognostic variables; in contrast, there are sex-related differences in clinical outcomes following imatinib therapy [29,30]. Evaluating dasatinib, although the STIM trial found that male sex was a positive predictive factor for treatment-free remission, most TKI discontinuation trials found that sex had no effect on this type of remission [31,32,33]. While female patients who stopped TKI therapy were predominately enrolled in the D-NewS Study, male and female patients were roughly equally represented in the first-line DADI and DASFREE studies [31,32,33].

Based on population pharmacokinetic studies, imatinib volume of distribution appears to be slightly influenced by age; it increases by 12% in people over 65 years, although this effect is not thought to be clinically significant [34]. Additionally, it has been demonstrated that limited clearance and low body weight are associated [35,36]. Thirty-one children participated in a phase I research study for the treatment of Ph+ leukaemia in youngsters [37]. The findings demonstrated that imatinib administered once daily to children produced plasma concentrations at a steady state and a mean area under the concentration that were equivalent to those observed in adults. We observed an inverse correlation between age and imatinib plasma concentration: drug levels decreased with an increasing age of the patients. On the contrary, Wilkinson and Larson reported that patients with elevated imatinib trough values were more likely to be 50 years of age or older, which is likely connected to an impaired liver or metabolism in older patients [38,39].

Eventually, we evaluated plasma and CSF concomitant samples at the Ctrough drug point. In comparison to other drugs, imatinib, dasatinib, nilotinib, and ponatinib showed the ability to cross the blood–brain barrier [40,41,42,43]. However, in our cohort, CSF sampling was not performed on the two patients undergoing nilotinib treatment. We observed that plasma dasatinib concentrations were significantly lower than those reported in CSF. Particularly, separating adult and paediatrics, the difference was retained only in children. The ability of dasatinib to substantially cross the blood–brain barrier is not universally agreed upon, and central nervous system reactions have been inconsistent [42,44,45]. Due to the quick metabolism of cytochrome P450 in the liver, only a very small drug amount is found in plasma. Targeted therapy blood–brain barrier penetration is difficult to predict; however, there are a few important physiochemical traits known to affect central nervous system penetration: the agent must be small, lipophilic, and without affinity for the main blood–brain barrier efflux pumps, such as p-glycoprotein (PGP) and breast-cancer resistance protein (BRCP). Dasatinib is a substrate of PGP and BCRP, small (506 g/mol), and lipophilic (consensus logP 2.8) [46]. Studies on dasatinib levels in CSF, however, were above all restricted to case reports [42,47,48,49,50].

## 5. Conclusions

Here, we describe the Ctrough concentration at the end of the dosing interval of four different TKIs, imatinib, dasatinib, nilotinib, and ponatinib in patients with CML. The data is presented also considering paediatrics (age less than 18 years) and adults and plasma and CSF samples separately. Analysing the obtained data, we can state that TDM could be a useful method for adjusting a patient’s treatment schedule depending on the drug concentrations in biological fluids together with other important tools as pharmacogenetics and the consideration of a patient’s age, gender, concomitant drugs, comorbidities, and so on. The use of TDM for TKIs for the treatment of CML is supported by many scientific studies in the literature. As a result, as the research develops, TKI TDM classification needs to be refined in terms of factors like sex and age. For these reasons, further studies collecting demographical, pharmacological, and genetic information are needed to confirm the observed results. Eventually, the limitations of our study need to be highlighted. First, the work lacks information regarding drug adverse events and clinical outcomes, data necessary to better underline the pharmacology of TKIs. In addition, the sample size is small and well-distributed, and the inclusion and exclusion criteria should be better defined.

Following the introduction of safe and effective targeted medications, more and more patients are experiencing positive results, which is accompanied by an increase in requests for dose reduction. The era of directing reduction with precision methods, such as TDM, and putting it into clinical practice in the near future to improve patients’ quality of life will dawn upon us when we expand the current knowledge with new studies in the area of personalized medicine.

## Figures and Tables

**Figure 1 biomedicines-11-02478-f001:**
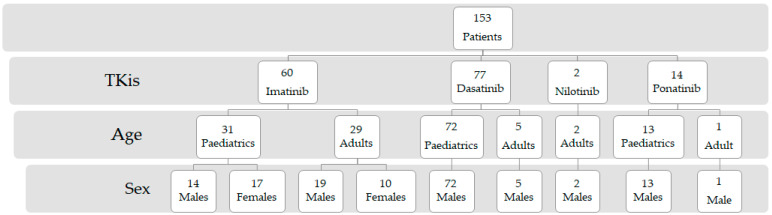
Flow chart of the patients’ distribution. The variables are represented as the frequency.

**Figure 2 biomedicines-11-02478-f002:**
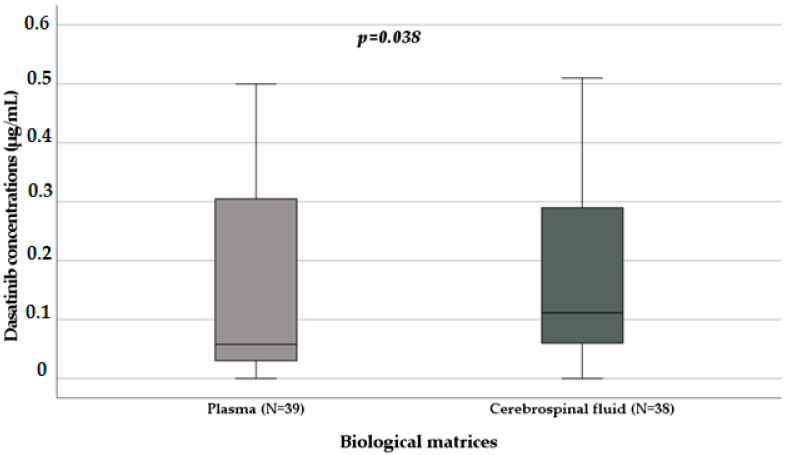
Distribution of dasatinib levels in plasma and CSF samples [μg/mL] (*p* = 0.038) obtained with the Mann–Whitney test. Box plot of dasatinib plasma concentration distribution in plasma and CSF samples; the boxes and black lines in the boxes represent, respectively, interquartile ranges (IQR) and median values. Median values (horizontal line), IQR (bars), patient values (black square), highest and lowest values (whiskers), and *p* value are shown.

**Figure 3 biomedicines-11-02478-f003:**
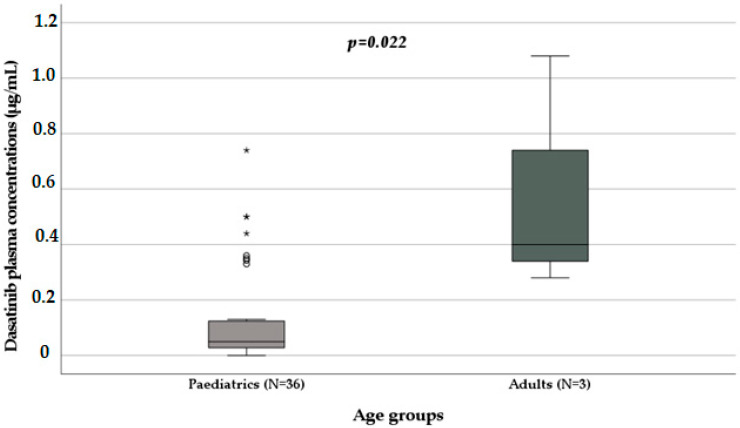
Influence of age on dasatinib plasma concentration [μg/mL] (*p* = 0.022) obtained with the Mann–Whitney test. Box plot of dasatinib plasma concentration distribution in paediatric and adult patients; the boxes and black lines in the boxes represent, respectively, interquartile ranges (IQR) and median values; the open dots and stars represent outlier values. Median values (horizontal line), IQR (bars), patient values (black square), highest and lowest values (whiskers), and *p* value are shown.

**Figure 4 biomedicines-11-02478-f004:**
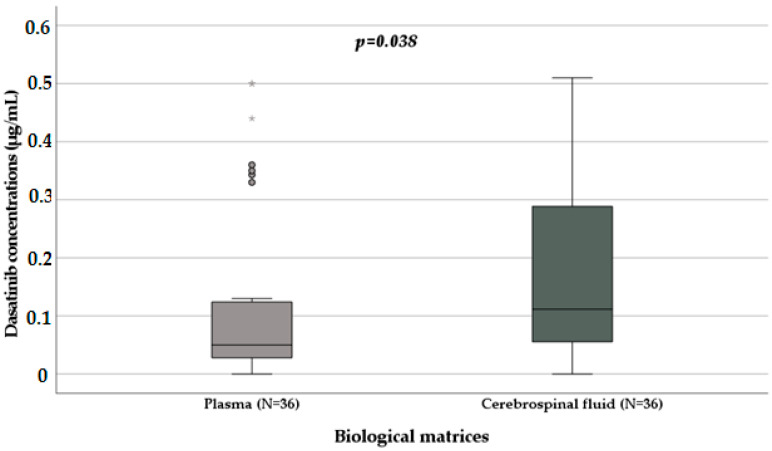
Distribution of dasatinib levels in plasma and CSF samples [μg/mL] (*p* = 0.038) considering paediatric patients obtained with the Mann–Whitney test. Box plot of dasatinib plasma concentration distribution in plasma and CSF samples; the boxes and black lines in the boxes represent, respectively, interquartile ranges (IQR) and median values; the open dots and stars represent outlier values. Median values (horizontal line), IQR (bars), patient values (black square), highest and lowest values (whiskers), and *p* value are shown.

**Table 1 biomedicines-11-02478-t001:** Demographics and pharmacological characteristics of the enrolled patients.

	All	Imatinib	Dasatinib	Nilotinib	Ponatinib
N	153	60	77	2	14
Males (N, %)	126, 82.4	33, 55	77, 100	2	14
Females (N, %)	27, 17.6	27, 45	0	0	0
Median age (years; IQR)	14 (12–17)	15 (25–56.75)	13 (12–17)	52 (50–54)	14 (13–16.5)
Paediatrics (N, %)	116, 75.8	31, 51.7	72, 93.5	0	13, 92.9
Median paediatric age (years; IQR)	13 (12–14)	13 (9–14)	13 (12–14)	/	14 (13–14)
Adults (N, %)	37, 24.2	29, 48.3	5, 6.5	2, 100	1, 7.1
Median adult age (years; IQR)	54 (42.5–58)	57 (50.5–59)	18 (18–48)	52 (50–54)	/
Plasma sampling (N, %)	111, 72.5	60, 100	39, 50.6	2, 100	10, 71.4
Plasma concentrations (µg/mL)	/	1.82 (0.88–2.93)	0.06 (0.03–0.33)	0.19 (0.16–0.22)	0.19 (0.15–0.51)
Cerebrospinal fluid sampling (N, %)	42, 27.5	0	38, 49.4	0	4, 28.6
Cerebrospinal fluid concentrations (µg/mL)	/	/	0.11 (0.06–0.32)	/	0.65 (0.18–1.65)

IQR: Interquartile range.

## Data Availability

The data are not publicly available due to ethical restrictions.

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
