# Peer review of "Pharmacokinetics of Four Tyrosine Kinase Inhibitors in Adult and Paediatric Chronic Myeloid Leukaemia Patients"

_biomedicines, 2023, doi:10.3390/biomedicines11092478_

Round 1

Reviewer 1 Report

The authors investigated the pharmacokinetics of TKI in CML patients. I have some comments.

1. The authors described the importance of TDM. However, the association of patients' adverse events, and efficacy with concentration of TKI. The authors should include the information of efficacy and adverse events. 

2. Figure2 and 3 are almost the same and confusing. Adult patients administered dasatinib are too small.  Do you want Figure4 ?

Author Response

Turin, August, 2023

Dear Editor,

Please, find enclosed a revised manuscript (Manuscript ID: biomedicines-2565728) to be considered for publication in Biomedicines, Special Issue on "Personalized Treatment in Cancer Research” as Article. As follow you can find the point-to-point response to reviewer questions. In addition, the uploaded version of the manuscript contains the highlighted correction (in yellow).

We hope that you will find our data worth of attention for Biomedicines readers.

Best regards,

Sarah Allegra

Reviewer 1

Comments and Suggestions for Authors

The authors investigated the pharmacokinetics of TKI in CML patients. I have some comments.

Dear Reviewer, thank you very much for the thorough review. We agree to all specific comments addressed and have revised our paper in light of the useful suggestions. Answers to the specific comments/suggestions/queries are as follows.

  1. The authors described the importance of TDM. However, the association of patients' adverse events, and efficacy with concentration of TKI. The authors should include the information of efficacy and adverse events.

Thank you for your professional review. We agree with you, it is necessary to evaluate adverse events and efficacy makers to better understand the pharmacokinetic of a drug, especially for those like TKIs. However, we do not have the possibility to include these data on our study and this is probably the major limitation of our study. Further works, including information about toxicity and clinical outcome are in planning stage. We insert this concept in the conclusion of our work.

  1. Figure2 and 3 are almost the same and confusing. Adult patients administered dasatinib are too small. Do you want Figure4 ?

Thank you for raising this point of potential confusion. The order of the figures has been modified and the choose to separately evaluate adults and paediatrics has been explained.

Reviewer 2 Report

Dear Editor and Authors,

It was a pleasure to review this manuscript titled “Pharmacokinetics of four tyrosine kinase inhibitors in adult and paediatric chronic myeloid leukaemia patients” by Dr. Allegra and her colleagues from the Laboratory of Clinical Pharmacology “Franco Ghezzo” at the University of Turin’s S. Luigi Gonzaga Hospital in Orbassano (TO), Italy.

In this prospective single institution study the authors attempt to evaluate plasma and CSF levels of a number or TKIs including  imatinib, dasatinib, nilotinib, and ponatinib in patients with chronic myeloid leukaemia. They then attempt to relate these to demographic factors such as age and gender.

This seems to be an interesting study with some limited interest to the clinical community but requires some editing and revision prior to acceptance. It also needs some language editing and better structuring of the presentation. Please see my comments below.

Comments:

1.       How were the patients in which CSF sampling was performed selected? What were the criteria and was there any possible bias?

2.       I don’t understand how the authors can claim this sampling (especially CSF) was routine clinical practice and did not require ethical approval (which by the way they did get!!). Please rewrite lines 87-89 to reflect this was an experimental practice!!

3.       Please explain more sections 2.2 and 2.3. Maybe section 2.5 should be moved up to 2.2! Why did you need to create stock solutions and to calibrates. You need to explain more the methods used to measure plasma and CSF levels in the patients and also explain that the plasma used to create the stock solutions were not of the study patients but blood bank donors (unrelated and non pathological!!).

4.        Was there a sample size calculation performed to select the 153 patients? Is this sample adequate to provide a statistically meaningful result? Was a power calculation performed after the conduct of the study? Why not?

5.       The authors need to explain the disparity in their groups. Certain TKIs have a higher number of pediatric patients while others have a balanced population. Is there an underlying reason for this disparity (i.e. approval of one drug for pediatric patients versus no approval? Or better efficacy?) Again how were the patient selection done? How were patients enrolled in the study? Was it random allocation, consecutive patients or other?

6.       Why was CSF sampling not performed in all patients?

7.       What are plasmatic levels? Do the authors mean plasma levels?

8.       Section 3.4 in the results section is too short and dry. Please explain more!

9.       We all know what TKIs are and how they work, so the first paragraph in the discussion is not needed and should be deleted. It seems to be added as filler for text length!

10.   The discussion lacks focus and structure. It seems to contain a lot of extraneous information (for example lines 214 – 241) which again it feels like it’s there to increase text length and offer nothing to the discussion!! Please address this issue!

11.   The conclusion needs to be toned down a little bit! TDM certainly has a role and will have a role in the future – especially if in the era of personalized medicine BUT it is not the Panacea and cure all the authors advocate!! It is a tool like all others we have available and this is how it should be presented (sensibly not ostentatious!!).

In conclusion, I need to see some major revising of this manuscript before I can feel happy to recommend its publication. Methodologically it seems to be adequate (although there is the issue of the sample size and selection bias) but the presentation and write up needs significant editing. By best regards to all and looking forward to the revision!

Needs some moderate editing!

Author Response

Turin, August, 2023

Dear Editor,

Please, find enclosed a revised manuscript (Manuscript ID: biomedicines-2565728) to be considered for publication in Biomedicines, Special Issue on "Personalized Treatment in Cancer Research” as Article. As follow you can find the point-to-point response to reviewer questions. In addition, the uploaded version of the manuscript contains the highlighted correction (in green).

We hope that you will find our data worth of attention for Biomedicines readers.

Best regards,

Sarah Allegra

Reviewer 2

Comments and Suggestions for Authors

It was a pleasure to review this manuscript titled “Pharmacokinetics of four tyrosine kinase inhibitors in adult and paediatric chronic myeloid leukaemia patients” by Dr. Allegra and her colleagues from the Laboratory of Clinical Pharmacology “Franco Ghezzo” at the University of Turin’s S. Luigi Gonzaga Hospital in Orbassano (TO), Italy.

In this prospective single institution study the authors attempt to evaluate plasma and CSF levels of a number or TKIs including imatinib, dasatinib, nilotinib, and ponatinib in patients with chronic myeloid leukaemia. They then attempt to relate these to demographic factors such as age and gender.

This seems to be an interesting study with some limited interest to the clinical community but requires some editing and revision prior to acceptance. It also needs some language editing and better structuring of the presentation. Please see my comments below.

Comments:

Dear Reviewer, thank you very much for the thorough review. We agree to all specific comments addressed and have revised our paper in light of the useful suggestions. Answers to the specific comments/suggestions/queries are as follows.

  1. How were the patients in which CSF sampling was performed selected? What were the criteria and was there any possible bias?

Thank you for your professional comment. We performed a monocentric cohort study in CML adults and paediatrics patients for which plasma or CSF monitoring was required to the Clinical Pharmacology Service “Franco Ghezzo” of the San Luigi Gonzaga University Hospital in Orbassano (Turin, Italy) from January 2017 and December 2022. Inclusion criteria were: CML diagnosis age above 2 years old and treated with dasatinib, imatinib, nilotinib or ponatinib for at least 6 months with a self-reported adherence of 90%. These information have been added in materials and methods section.

  1. I don’t understand how the authors can claim this sampling (especially CSF) was routine clinical practice and did not require ethical approval (which by the way they did get!!). Please rewrite lines 87-89 to reflect this was an experimental practice!!

Thank you for raising this point of potential confusion. There was an error and the text has been changed as follows: Calibration curve and QCs were routinely recorded during daily clinical practice as a quality assurance measure and in order to explore improvements in the quality of ser-vices.

  1. Please explain more sections 2.2 and 2.3. Maybe section 2.5 should be moved up to 2.2! Why did you need to create stock solutions and to calibrates. You need to explain more the methods used to measure plasma and CSF levels in the patients and also explain that the plasma used to create the stock solutions were not of the study patients but blood bank donors (unrelated and non pathological!!).

Thank you for the revision. As you suggested the section has been revised:

2.2. Chemicals

Imatinib, nilotinib dasatinib, ponatinib, triethylamine, methanol and acetonitrile were purchased from Sigma-Aldrich (Milan, Italy). A Milli DI system from Millipore (Milan, Italy) were used to create High Pressure Liquid Chromatography (HPLC)-grade water. For the chromatographic run has been used a mobile phase constituted by a 40% of solution A (72.5% H2O + 25% CH3OH + 2.5% C6H15N), 40% C2H3N, 20% CH3OH.

2.3. Stock solutions, calibration standards (STDs), quality controls (QCs), plasma patients samples.

The Blood Bank of S San Luigi Gonzaga University Hospital in Orbassano (Turin, Italy) generously provided blank plasma from healthy donors used to prepare STDs and QCs. Drug stock solutions were created in methanol at a final concentration of 1 mg/mL and kept at -4°C for no longer than three months. The internal standard (IS) for dasatinib, imatinib and ponatinib evaluations was nilotinib, whereas for the nilotinib evaluation, the IS was imatinib. The 50 g/mL methanol-based IS solutions were prepared and utilized right away. The highest calibration standard (STD8) and the highest quality control (QC5) were made by diluting donors’ blank plasma serially with the other STDs and QCs, and the highest calibration standard (STD8) and QC5 were made by adding a specific volume of stock solutions to blank plasma. All drugs used the same calibration range and QC concentrations (STDs: 0.005, 0.01, 0.05, 0.1, 0.5, 1, 5, 10 µg/mL; QCs: 0.05, 0.5, 5 µg/mL). Prior to analysis, the STDs and QCs were kept at -4°C for no more than three months, avoiding multiple freeze-thaw cycles. Calibration curve and QCs were routinely recorded during daily clinical practice as a quality assurance measure and in order to explore improvements in the quality of services. The limit of detection (LOD) in plasma was defined as the concentration referred to a signal-to-noise ratio of 3/1; the lowest concentration levels determined with a percent of deviation from the nominal concentration, and a relative standard deviation < 20%, was considered the lowest limit of quantification (LOQ).

Plasma samples, from CML patients undergoing imatinib, dasatinib, nilotinib or ponatinib treatment, were obtained after blood samples centrifugation at 1,500 g for 10 minutes at 4°C and kept at -4°C till HPLC coupled with ultraviolet detection (UV) analysis. CSF samples, collected by an experienced physician in the lumbar region of CML patients undergoing imatinib, dasatinib, nilotinib or ponatinib treatment, were directly frozen till HPLC-UV analysis was performed.

2.4. Sample preparation

Patients plasma and CSF samples, STDs and QCs were all treated with the following procedure: firstly, for dasatinib and ponatib, 50µl of IS were added to 500µl of sample. Then, 500µl of C2H3N were used for the deproteinization and vortexed for 30 seconds be-fore carrying out the centrifugation (15’ at 12.000rpm). The following steps need the usage of a void pump: C18 Solid Phase Extraction (SPE; Grace, Italy) columns were conditioned with 1ml of CH3OH and 1ml of H2O; 800µl of the obtained sample supernatant has been transferred onto the SPE columns, with 1ml of H2O. After removing the waste, 500µl of CH3OH was added, therefore the solution needed to evaporate until a pellet was obtained. Finally, 250µl of mobile phase was added: the solution was then ready for the HPLC-UV analysis. Differently, for nilotinib and imatinib, 50µl of IS were added to 500µl of plasma or CSF samples. Then, 750µl of extracting solution (C2H3N:CH3OH 50:50) has been added and vortexed for 30 seconds before carrying out the centrifugation (10’ at 12.000rpm). The solution was then ready for the HPLC-UV analysis.

2.5 Chromatographic system and conditions

The chromatographic analysis for the intracellular quantification of the four drugs has been carried out using a HPLC-UV system. HPLC was performed with the Agilent 1100 system (Santa Clara, United States) equipped with an autosampler, a spectrophotometer, and a heated column compartment. System management and data acquisition were performed with the ChemStation Agilent software Santa Clara, United States). The chromatographic separation has been realized at 40°C using a HyPURITY C18 (Thermo-Scientific, Monza, Italy) 150x4.6mm 3µ column. The analysis was carried out at the constant flow rate of 0.9 mL/min at 35°C in an isocratic condition. The eluate was monitored at 267 nm. The time of the analytical run was 10 minutes, according to the retention times of the substances and their good separation.

  1. Was there a sample size calculation performed to select the 153 patients? Is this sample adequate to provide a statistically meaningful result? Was a power calculation performed after the conduct of the study? Why not?

Thank you for your suggestion. All the variables were tested for normality with the Shapiro–Wilk test. The correspondence of each parameter was evaluated with a normal or non-normal distribution, through the Kolmogorov–Smirnov test. Firth’s correction was applied to reduce the bias of the estimates due to small number of events. We added this information in Statistical analysis section.

  1. The authors need to explain the disparity in their groups. Certain TKIs have a higher number of pediatric patients while others have a balanced population. Is there an underlying reason for this disparity (i.e. approval of one drug for pediatric patients versus no approval? Or better efficacy?) Again how were the patient selection done? How were patients enrolled in the study? Was it random allocation, consecutive patients or other?

Thank you for your comment. A more informative detail about patient’s enrolment have been added in 2.1 section. Moreover, the possible criteria regarding the choice of treatment have been inserted in discussion section.

  1. Why was CSF sampling not performed in all patients?

Thank you for your interest on our work. The choice of physician to perform the CFS sampling was related to the involvement of CNS of the enrolled patients.

  1. What are plasmatic levels? Do the authors mean plasma levels?

Sorry for the mistake, the text has been corrected according to your advice.

  1. Section 3.4 in the results section is too short and dry. Please explain more!

Thank you for your comment. All the results section has been revised and expanded.

  1. We all know what TKIs are and how they work, so the first paragraph in the discussion is not needed and should be deleted. It seems to be added as filler for text length!

Thank you for your suggestion, the first paragraph has been removed and substituted with the explanation of the criteria used for TKIs treatment choice.

  1. The discussion lacks focus and structure. It seems to contain a lot of extraneous information (for example lines 214 – 241) which again it feels like it’s there to increase text length and offer nothing to the discussion!! Please address this issue!

Thank you for your revision. The first part of the discussion section has been completely revised.

  1. The conclusion needs to be toned down a little bit! TDM certainly has a role and will have a role in the future – especially if in the era of personalized medicine BUT it is not the Panacea and cure all the authors advocate!! It is a tool like all others we have available and this is how it should be presented (sensibly not ostentatious!!).

Thank you for your interest on our work. As you suggested, the conclusion section has been completely revised as follows:

Here we describe the Ctrough concentration at the end of dosing interval of four different TKIs, imatinib, dasatinib, nilotinib and ponatinib, in CML patients. The data have been presented also considering paediatrics (with less the 18 years old) and adults, and plasma and CSF samples separately. Analysing the obtained data, we can state that TDM could be a useful method for adjusting a patient treatment schedule depending on drug concentrations in biological fluids, together with other important tools as pharmaco-genetics and the consideration of patients information as age, gender, concomitant drugs, comorbidities and so on. The use of TDM for TKIs for the treatment of CML is supported by many literature scientific evidences. As a result, as the research develops, TKIs TDM classification needs to be refined also in terms of factors like sexes and ages. For these rea-sons, further studies collecting demographical, pharmacological and genetic information are needed to confirm the observed results. Eventually, the limitations of our study need to be highlighted. First, the work lacks of information regarding drugs adverse events and clinical outcome, data necessary to better underline the pharmacology of TKIs. Then, the ample size need is small and well distributed and inclusion and exclusion criteria should be better defined.

Following the introduction of safe and effective targeted medications, more and more patients are experiencing positive results, which is accompanied by an increase in re-quests for dose reduction. The era of directing reduction by precision methods, as TDM, and putting it into clinical practice in the near future to improve patients quality of life will dawn upon us when we expand the current knowledge with new studies in the area of personalized medicine.

In conclusion, I need to see some major revising of this manuscript before I can feel happy to recommend its publication. Methodologically it seems to be adequate (although there is the issue of the sample size and selection bias) but the presentation and write up needs significant editing. By best regards to all and looking forward to the revision!

Round 2

Reviewer 1 Report

The manuscript is adequately revised.

Author Response

Dear Reviewer, thank you very much for the thorough review.

Reviewer 2 Report

Dear Editor and Authors,

Thank you for asking me to re-review this manuscript by Dr. Allegra and her colleagues titled “Pharmacokinetics of four tyrosine kinase inhibitors in adult and paediatric chronic myeloid leukaemia patients”.

The authors have made some efforts to revise their manuscript, however there are still issues that I raised in my initial review that have not been addressed!

Specifically:

1.       My question regarding the performance of a sample size calculation or a power analysis was circumvented and not addressed!

2.       My question about the disparity between their study groups, especially amongst pediatric patients and certain TKIs has not been adequately addressed!! The authors need to further discuss this disparity!

3.       My suggestion regarding re-arranging the subsections in the material and methods section was not heeded!! I still think talking about how testing was performed should be presented first and subsequently the chemical solutions ect!!

The rest of my comments have been addressed well.

Thanks for the effort.

Needs some minor language editing.

Author Response

Dear Editor,

Please, find enclosed a revised manuscript (Manuscript ID: biomedicines-2565728) to be considered for publication in Biomedicines, Special Issue on "Personalized Treatment in Cancer Research” as Article. As follow you can find the point-to-point response to reviewer questions. In addition, the uploaded version of the manuscript contains the highlighted correction (in green).

We hope that you will find our data worth of attention for Biomedicines readers.

Best regards,

Sarah Allegra

Reviewer 2

Dear Editor and Authors,

Thank you for asking me to re-review this manuscript by Dr. Allegra and her colleagues titled “Pharmacokinetics of four tyrosine kinase inhibitors in adult and paediatric chronic myeloid leukaemia patients”.

The authors have made some efforts to revise their manuscript, however there are still issues that I raised in my initial review that have not been addressed!

Dear Reviewer, thank you very much for the thorough review. We agree to all specific comments addressed and have revised our paper in light of the useful suggestions. Answers to the specific comments/suggestions/queries are as follows.

Specifically:

  1. My question regarding the performance of a sample size calculation or a power analysis was circumvented and not addressed!

Thank you for your revision. The one-way ANOVA test has been performed to calculate the power analysis, as reported in 2.6 and 3.1 sections.

  1. My question about the disparity between their study groups, especially amongst pediatric patients and certain TKIs has not been adequately addressed!! The authors need to further discuss this disparity!

Thank you for your suggestion. Possible explanation of distribution disparities had been reported in lines 335-343.

  1. My suggestion regarding re-arranging the subsections in the material and methods section was not heeded!! I still think talking about how testing was performed should be presented first and subsequently the chemical solutions ect!!

Dear Reviewer, the subsections in materials and methods have been reorganized, as you suggested.

The rest of my comments have been addressed well.

Thanks for the effort.

Thank you for your time and effort in revising our work.
